# Instructed Language Models with Retrievers Are Powerful Entity Linkers

**Zilin Xiao**♠  **Ming Gong**♣  **Jie Wu**♣
**Xingyao Zhang**♣  **Linjun Shou**♣  **Daxin Jiang**♣
Rice University♠  Microsoft STCA♣
zilin@rice.edu
{migon, jiewu1, xingyaozhang, lisho, djiang}@microsoft.com

## Abstract

Generative approaches powered by large language models (LLMs) have demonstrated emergent abilities in tasks that require complex reasoning abilities. Yet the generative nature still makes the generated content suffer from hallucinations, thus unsuitable for entity-centric tasks like entity linking (EL) requiring precise entity predictions over a large knowledge base. We present Instructed Generative Entity Linker (INSGENEL), the first approach that enables casual language models to perform entity linking over knowledge bases. Several methods to equip language models with EL capability were proposed in this work, including (i) a sequence-to-sequence training EL objective with instruction-tuning, (ii) a novel generative EL framework based on a light-weight potential mention retriever that frees the model from heavy and non-parallelizable decoding, achieving 4× speedup without compromise on linking metrics. INSGENEL outperforms previous generative alternatives with +6.8 F1 points gain on average, also with a huge advantage in training data efficiency and training compute consumption. In addition, our skillfully engineered in-context learning (ICL) framework for EL still lags behind INSGENEL significantly, reaffirming that the EL task remains a persistent hurdle for general LLMs.

## 1 Introduction

Entity linking (EL) has emerged as a critical research problem in the intersection of Natural Language Processing (NLP) and Information Retrieval (IR), and it serves as a fundamental task in various NLP applications, including text understanding (Févry et al., 2020), question answering (Asai et al., 2020), to name a few.

Previous EL approaches typically divided EL into two steps: Mention Detection (MD) and Entity Disambiguation (ED). Once the MD model produces mention span proposals over the input document, dense ED models attempt to encode mention context and entity candidates into representations. Then a simple maximum inner product search (MIPS) is utilized to capture mention-entity correspondence, resulting in final entity predictions over the entire knowledge base (KB).

A more recent trend in EL research focuses on building an end-to-end pipeline that intertwines MD and ED and formulates them into different tasks, such as question-answering (Zhang et al., 2022b), multi-task learning (Ayoola et al., 2022) and language generation (Cao et al., 2021b).

While generative large language models (LLMs) have shown dominant abilities in a multitude of NLP tasks (Wang et al., 2022; Ouyang et al., 2022; Xu et al., 2023), their abilities are under-explored in the realm of entity-centric NLP tasks, especially EL. Unlike numerous knowledge language grounding tasks that can be readily unified to text-to-text frameworks (Xie et al., 2022), thanks to their close adherence to a unified surface form (*i.e.*, text), EL presents distinctive challenges. The foremost difficulty lies in the fact that unconstrained generation frequently fails to yield precise entity identifiers, because of the notorious hallucination issue of LLMs. Even though the pre-defined prefix trie proposed in Cao et al. (2021b) can ensure a legal generation sequence, the non-parallel beam search process makes it unsuitable for real-time usage, not to mention its performance lags behind later discriminative alternative. In this work, we revisit generative EL by proposing three variants: INS-GENEL, INSGENEL-R and INSGENEL-ICL.

INSGENEL solves EL by leveraging a methodology which constrains the next possible tokens, and eliminates invalid options during generation, thus ensuring the generated text can be successfully parsed into EL results. In contrast to Cao et al. (2021b), the baseline model in INS-GENEL diverges by incorporating the input document as part of the prompt and optimizes the casual language modeling (CLM) objective on

decoder-only transformer models through instruction fine-tining approach, as opposed to employing a sequence-to-sequence objective based on an encoder-decoder neural architecture. We observe that by instruction fine-tuning decoder-only LMs such as OPT-series (Zhang et al., 2022a) and LLaMA-series (Touvron et al., 2023) , INSGENEL yields superior EL performance compared to previous work that finetunes a generative alternative on BART (Lewis et al., 2020a) with +6.8 F1 points on average. This suggests that instruction fine-tuning may unlock specific entity-related knowledge within pretrained language models. Additionally, this approach exhibits significant improvements in both training compute efficiency and data efficiency, indicating that foundation language models can effectively reduce the learning difficulty of task-specific objectives.

However, directly generating sequences incurs significant computational overhead during inference, as both memory footprint and computation increase with sequence length, not to mention the non-parallelizable nature of auto-regressive decoding. To address these challenges, we propose offloading the Mention Detection (MD) responsibility to an external retriever. For each document, the external retriever selects top-$K$ entities that possibly exist in the document and constructs a possible mention set. Then the surface form matching process dynamically determines the range where decisions are needed during the generation. Finally, greedy decoding is employed only when a choice is necessary (within a decision range), $i.e.$, whether to start a mention, end an entity identifier, or choose among dynamic candidates.

Named INSGENEL-R, this novel EL generation framework offers several key advantages: a) Compared to constrained beam search, INSGENEL-R significantly reduces the number of heavy generation forwards at the cost of a simple vector retrieval. b) It does not fall into beam failure, $i.e.$, getting stuck with improbable mentions during the process of generating mention spans, thereby wasting valuable inference compute. c) It is less likely to miss obvious mentions, while traditional generative EL tends to make mistakes when generating mention boundaries. INSGENEL-R achieves $4\times$ reduction in terms of the number of LM calls, reduces runtime by a similar ratio at the minor cost of performance decline. Moreover, we extend the usage of the same retriever in an in-context learning (ICL)

manner on LLMs such as GPT-3 (Brown et al., 2020), GPT-3.5 and Codex, named INSGENEL-ICL. Side-by-side comparative results indicate that while generic LLMs can correctly adhere to the format of exemplars by learning in context, they are unable to match the same accuracy exhibited by INSGENEL-R.

In summary, this paper pushes the generative EL paradigm to new limits, both in terms of evaluation metrics and runtime performance. The novel retrieval-augmented generative EL emulates an agent interacting with the dynamic environment, and sheds light on real-time generative EL without incurring a substantial metric drop. The new in-context learning paradigm for EL also indicates that current LLMs can not support an optimal ICL solution for EL. We release our code and checkpoints at `https://github.com/MrZilinXiao/InsGenEntityLinking`.

## 2 Related Works

**Entity Linking (EL)** is a task of locating mentions and disambiguating these surface forms into entities in some knowledge base. Each EL mention-entity prediction should be in the format of $\langle m_s, m_e, ent \rangle$, where $m_s, m_e, ent$ indicate the start, end position of a mention and the entity identifier in the knowledge base respectively.

While early EL methods (Hoffmann et al., 2011; Daiber et al., 2013) treat EL as decomposed subtasks, such as Mention Detection (MD) and Entity Disambiguation (ED), a more recent trend is considering EL an end-to-end task. Kolitsas et al. (2018) first propose to use a neural-based model to conduct MD and ED jointly. Broscheit (2019) transform the EL into a BIO tagging problem by training a token-classification model with an external entity classification head. Zhang et al. (2022b) formulate EL into a question-answering (QA) problem and borrowed the popular retrieve-then-read pipeline in QA. Ayoola et al. (2022) leverage the type and description of entities and employs aggregation of discriminative scores to obtain the final result.

For the generative EL paradigm, Cao et al. (2021b) first turn EL into sequence-to-sequence constrained generation with an encoder-decoder transformer model, fully exploiting the flexibility of sequence generation. Cao et al. (2022) extend the same paradigm into multilingual EL. De Cao et al. (2021) propose an efficient generative EL

model that relies on a shallow LSTM-based decoder, at the cost of generalization over general EL benchmarks. Mrini et al. (2022a) add an entity-matching prediction module over generated sequence and train EL using the MD objective, autoregressive objective and entity-matching objective jointly.

Our methods fall within the generative EL category as well, but they stand out in several distinctive features. First, putting the input document in the instruction enables EL generation on decoder-only casual language models rather than using an encoder to capture the global context. This approach enables us to benefit from the advancements in recent foundation models. Moreover, the mention detection offloading alleviates the burden of non-parallel generation. The practice of invoking the generative model only within the decision range drastically enhances computation efficiency, also presenting an intriguing parallel with interactive agents in knowledge base question answering (KBQA) (Gu et al., 2022) and the principle of Interactive NLP (Wang et al., 2023).

**Instruction-tuning** (Wei et al., 2022a) usually refers to finetuning language models with a collection of tasks that are formulated as plain-text instructions. Recent LLM pre-training (Brown et al., 2020; Chowdhery et al., 2022) show that emergent abilities (Wei et al., 2022b) exhibit when model size, data and training compute scale. Specifically, one of such abilities is that large models can leverage natural language instructions to solve language problems in a zero-shot fashion. When instructions alone can not guide satisfactory generation, **In-Context Learning (ICL)** can guide LLM to learn from in-context exemplars to perform complex reasoning, such as solving mathematical problems (Wei et al., 2022c).

Although instruction-tuning was originally proposed for zero-shot language tasks, we find that tuning models using document instructions also leads to improved prediction and disambiguation of uncommon entities in generative EL. Taking inspiration from ICL, we have extended our approaches to encapsulate EL with the ICL paradigm, facilitating an equitable comparison between INSGENEL-R and general-purpose LLMs.

**Retrieval-augmented Language Models** are an emerging class of models in the field of NLP. They offer innovative solutions to knowledge-related tasks by combining the power of language mod-els with the ability to retrieve relevant information from a large corpus of external knowledge. Most works augment the input context with retrieved external documents, thus the encoded representation or generated sequence will be conditioned on external knowledge. Guu et al. (2020) firstly train an end-to-end language encoder with a dense retrieval system. Lewis et al. (2020b) finetune a general sequence-to-sequence model with an external retriever.

Our INSGENEL-R utilizes a lightweight encoder for each document to retrieve the most relevant entities, then uses a generative agent to eliminate impossible solutions dynamically. For similar entity-related retrieval scenarios, Févry et al. (2020) is the first to integrate entity supervision on language understanding tasks. Zhang et al. (2022c) constructs an entity memory bank and allows dynamic aggregation of entity representations, boosting the performance of entity-intensive question-answering and generation tasks. However, none of them use retrieved entities to explicitly guide text generation.

## 3 Methodology

### 3.1 Vanilla Generative EL

Vanilla generative EL addresses entity linking as an autoregressive sequence generation task, that is, given the document, the generated sequence should both indicate the mentions and their associated KB entities. Special boundary tokens are used to mark up mentioned plain string and entity identifiers. The training setup generally follows a standard sequence-to-sequence neural machine translation task (Wu et al., 2016) , where Cao et al. (2021b) maximize $p_\theta(y \mid x)$ with respect to the model's parameters $\theta$. We refer to Appendix B for inherent problems of vanilla generative EL.

### 3.2 Instruction-tuned INSGENEL Baseline

Our baseline focuses on instruction-tuning a decoder-only casual language model. The prompt part consists of an optional natural language task instruction plus the document to be linked. The target sequence comprises the linked document in its plain-text form, but with special boundary symbols indicating mentions and corresponding entity identifiers[1].

---

[1]Following Cao et al. (2021b), we use parentheses as mention boundaries and brackets as entity identifier boundaries.

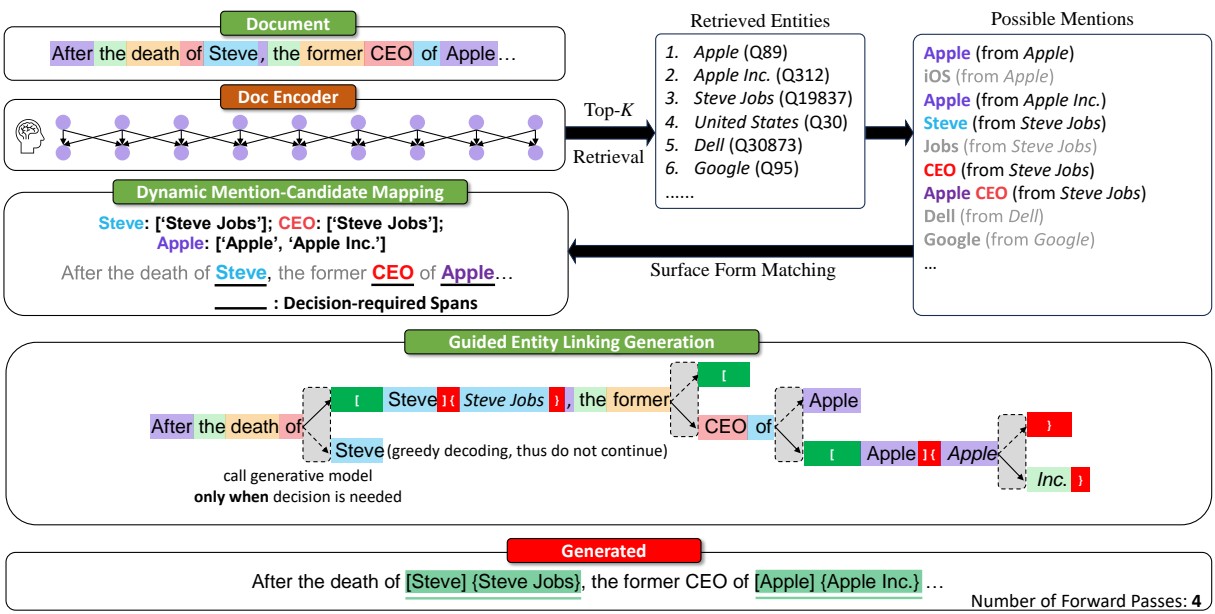

Figure 1: Overview of INSGENEL-R with greedy decoding strategy. Each box with grey background indicates a generative agent decision, and the dotted arrow denotes an abandoned decoding path. Best viewed in color and be in comparison with Appendix B and Figure 4.

We optimize the following cross-entropy loss, which is known as the next token prediction loss in casual language modeling:

$$\mathcal{L}_{\mathrm{EL}} = -\sum_{i=n}^{N} \log P\left(y_i \mid y_1, \ldots, y_{i-1}\right), \quad (1)$$

where $\mathbf{y} = [y_1, \ldots, y_n, \ldots, y_N]$ is the concatenation of the prompt and target sequence, and $n$ denotes the length of the prompt sequence. Note that we do not backward the next token prediction loss on the prompt sequence. The fine-tuned model will serve as the generative backbone of all experiments in this paper.

### 3.3 INSGENEL-R: Retrieval-augmented Generative EL

Given a document chunk $x \in \mathcal{X}$, we would like to build a dual encoder that retrieves top-$K$ candidate entities $\{e_1, e_2, e_3, \ldots, e_k\}$ which might be mentioned in $x$. The retriever computes document representations $X^p$ and entity representations $E^e$ as follows,

$$X = \mathbf{BERT}_P\left([\texttt{CLS}]; x; [\texttt{SEP}]\right),$$
$$E^e = \mathbf{BERT}_E\left([\texttt{CLS}]; \phi_{\mathrm{title}}(e); \phi_{\mathrm{desc}}(e); [\texttt{SEP}]\right),$$

where $\mathbf{BERT}_P$ and $\mathbf{BERT}_E$ are two BERT (Devlin et al., 2019) text encoders that do not share

weights, [CLS] and [SEP] are BERT special tokens. $\phi_{\mathrm{title}}(e)$ and $\phi_{\mathrm{desc}}(e)$ are text identifier and text description of an entity $e$, respectively.

Specifically, we use a multi-label variant of noise contrastive estimation (NCE) (Gutmann and Hyvärinen, 2010) objective to train an entity retriever conditioned on document input, following the setup in Zhang et al. (2022b).

During training, we prepare a document chunk $x$ and a set of oracle entities $\mathcal{E}(x) \in \mathcal{E}$ that are mentioned in $x$. We train the retriever with maximizing the following objective:

$$\sum_{e \in \mathcal{E}(x)} \log\left(\frac{\exp\left(S(e)\right)}{\exp\left(S(e)\right) + \sum_{e' \in \mathbf{N}(\mathcal{E}, x)} \exp\left(S(e')\right)}\right)$$

where $S(e) = X_1^\top E_1^e$ stands for the matching score between document chunk $x$ and entity $e$, $\mathbf{N}(\mathcal{E}, x)$ is a set of negative entities that do not overlap with gold entity set $\mathcal{E}(x)$. This objective constructs NCE instances on the fly, treating one gold entity as the only correct answer in each training sample, while excluding other gold entities out of negative examples. 90% negative samples are selected randomly and 10% are chosen by hard negative mining, *i.e.*, selecting the highest-scoring incorrect entity.

During inference, entity representations $E^e$ are cached into Faiss (Johnson et al., 2021) index to al-

low fast top-$K$ retrieval. With retrieved entities, we construct a set of possible mentions by looking up an entity-to-mention dictionary[2]. The top-right corner of Figure 1 illustrates an example set of possible mentions. Each tuple within the possible mention set comprises one of the $k$ entities retrieved and its associated mention string. Be aware that several different entities can correspond to the same mention string.

Then, we run surface form matching between a possible mention set and document text. Any parts of document text that match possible mentions are marked as decision-required[3]. Each decision-required span comprises the start and end indices, and possible mentions that may be within the span.

In the Guided Entity Linking Generation stage, the generative agent will determine the next action based on its current state:

1. Out of a decision-required span: Unlike Vanilla Generative EL in 3.1, which needs to decide whether to initiate a mention prediction at each document token, INSGENEL-R only needs to directly copy the next token when out of a decision-required span.

2. At the beginning of a decision-required span: INSGENEL-R has to decide when to start a mention within a decision-required span. This is achieved by comparing the log probability of next document token and mention start boundary token. A constant score offset is added to the mention start token due to the elevated probability of a mention appearing within a decision-required span. Note that it is also a valid choice for a decision-required span not to generate any mention at all, like the "CEO" span in Figure 1.

3. Within the mention part of a decision-required span: Once a mention has been initiated, if there is only one possible mention with this span, the agent will directly copy this mention (as in the case of "Steve" in sky-blue font in Figure 1). If not, a decision is made on which mention to choose within this span, which is constrained by a dynamically generated prefix tree that covers all mention choices in the span.

4. Within the entity part of a decision-required span: Once a span has completed the decoding of the mention, the agent will continue to decode the entity identifier part. Similar to the decoding of the mention part, if there is only one entity associated with the decoded mention, the agent will directly copy this candidate entity (such as "Steve Jobs" with italic font in Figure 1). Otherwise, the agent will dynamically construct a prefix tree containing associated entities to constrain the generation of the entity identifier (such as "Apple" and "Apple Inc." in italics in Figure 1).

Incurring only the cost of one dense vector retrieval, INSGENEL-R reduces the calls to the generative model by 90% in this sample document and no longer relies on a massive, pre-defined prefix tree. Since the retrieval procedure takes into account the entity description, it mitigates the challenge inherent to the generative EL paradigm, that is to distinguish between entities with similar identifiers.

### 3.4 INSGENEL-ICL: In-Context Learning Entity Linking Paradigm

In-context learning (ICL) with large language models (LLMs) demonstrates strong zero-shot and few-shot performance in many NLP tasks. However, the direct application of ICL to entity linking (EL) is difficult, primarily due to the limitations on the context window size which prevent the generative model from directly accessing the overwhelming number of candidate entity identifiers. Nonetheless, equipped with a well-trained retriever in INSGENEL-R, we condense the EL task into an advanced machine reading comprehension (MRC) problem: given potential entities and a document, the LLM is required to choose the mention span and the respective entity from a document.

The INSGENEL-ICL paradigm begins with a fixed exemplar and task instruction, both of which are fed to the LLM as an in-context demonstration. The task instruction prompt words have been iteratively refined, integrating well-known engineering techniques for prompting such as bad case demonstrations, and have leveraged automatic prompt optimization tricks. We encourage readers to Figure 5 in the Appendix for the in-context prompt template.

Notably, each prediction will have its final result matched by a regular expression; to prevent failed parsing due to multiple identical surface forms

---

[2]We refer readers to Appendix C.2 for details of building such a dictionary.

[3]Note that the marked segments may overlap, and will require merging mentions into a unified decision span through the algorithm provided in Appendix C.4.

| Category | Method | In-domain | | Out-of-domain | | | | | | | Avg |
|---|---|---|---|---|---|---|---|---|---|---|---|
| | | AIDA | MSNBC | Der | K50 | R128 | R500 | OKE15 | OKE16 | |
| Discriminative | Hoffart et al. (2011) | 72.8[*] | 65.1 | 32.6 | 55.4 | 46.4 | 42.4 | 63.1 | 0.0 | 47.2 |
| | Kolitsas et al. (2018) | 82.4[*] | 72.4 | 34.1 | 35.2 | 50.3 | 38.2 | 61.9 | 52.7 | 53.4 |
| | van Hulst et al. (2020) | 80.5[*] | 72.4 | 41.1 | 50.7 | 49.9 | 35.0 | 63.1 | 58.3 | 56.4 |
| | Zhang et al. (2022b) | **85.8**[*] | 72.1 | 52.9 | 64.5 | 54.1 | 41.9 | 61.1 | 51.3 | 60.5 |
| | Ayoola et al. (2022) | 84.0[*] | 71.8 | 50.7 | 64.7 | 58.1 | 42.0 | 64.4 | 59.1 | 61.9 |
| Generative | Cao et al. (2021b) | 83.7[*] | 73.7 | 54.1 | 60.7 | 46.7 | 40.3 | 56.1 | 50.0 | 58.2 |
| | Cao et al. (2021a) | 85.5[*] | - | - | - | - | - | - | - | - |
| | Mrini et al. (2022b) | 85.7[*] | - | - | - | - | - | - | - | - |
| Ours | INSGENEL | 81.5 | 69.5 | **60.9** | **73.8** | **58.6** | **46.8** | 65.7 | 62.1 | **64.9** |
| | INSGENEL-R | 80.6 | **74.2** | 59.8 | 71.9 | 56.8 | 45.5 | 64.1 | **63.3** | 64.5 |

Table 1: *InKB* Micro F$_1$ on the eight popular test sets. For each dataset, **bold** indicates the best model and underline indicates the second best. Metric with [*] denotes that this model trains on the AIDA-CoNLL train split, while our methods do not utilize any in-domain train set. - indicates the authors neither report the metric on certain test sets nor release their code and checkpoints.

appearing in the same document, we require the model to output not only the mention text in the exemplar, but also the surrounding context for precise span matching.

Considering that the inputs for INSGENEL-R and INSGENEL-ICL are entirely identical, INSGENEL-ICL can serve as a fair comparison point for in-context learning in LLM, which helps examine the distinctions between a generic LLM and a fine-tuned generative model when performing generative EL.

## 4 Experiment

### 4.1 Setting

**Datasets.** We follow the established standard and report *InKB*[4] Micro F1 score on eight entity linking datasets. Specifically, we use eight out-of-domain test sets: the AIDA-CoNLL (Hoffart et al., 2011) test split, MSNBC (Cucerzan, 2007), Derczynski (Der) (Derczynski et al., 2015), KORE 50 (K50) (Hoffart et al., 2012a), N3-Reuters-128 (R128), N3-RSS-500 (R500) (Röder et al., 2014), and OKE challenge 2015 and 2016 (OKE15 and OKE16) (Nuzzolese et al., 2015). Training datasets were built from all article abstracts from English Wikipedia 2023-02-20 dump. Notably, we do not fine-tune our models on domain-specific datasets, but rely solely on Wikipedia, to examine the generalization capability of our method. This means we do not use the train split of the AIDA-CoNLL dataset. We use hyperlinks in Wikipedia as entity

---

[4]Following common practice in previous works (Ayoola et al., 2022; Zhang et al., 2022b; Kolitsas et al., 2018), only mentions with valid KB entities are used for evaluation.

labels with a string matching heuristic to solve coreference following Cao et al. (2021a), because when an entity is mentioned multiple times in a Wikipedia article, often only the first mention is correctly linked. Additionally, we construct weak entity labels to increase Wikipedia data quality according to Ayoola et al. (2022).

**Training and Evaluation.** We utilize two series of decoder-only models as our base models: LLaMA (Touvron et al., 2023) and OPT (Zhang et al., 2022a). The OPT series provide pre-trained models of varying sizes, enabling us to examine the correlation between model size and generative EL performance. The best result is reported on LLaMA 7B version. We do not conduct hyperparameter search; the hyperparameters used during training are detailed in Appendix A. All pieces of training were performed on a node with 8 V100-SXM2-32GB GPUs, utilizing DeepSpeed (Rasley et al., 2020) for distributed training management. We train all models for one epoch. We report training time, size of training data and training compute in Section 4.3. The evaluation was conducted on ELEVANT (Bast et al., 2022) platform with a single V100 GPU. Best-performing INSGENEL-R is with $k = 100$, and we report the impact of $k$ in Section 4.3.

### 4.2 Main Result

We report the model evaluation results in Table 1. Our model exhibits consistent performance advantages across all test sets, excluding AIDA. This achievement is noteworthy given that, unlike all preceding works, we did not apply domain-

specific fine-tuning on AIDA. Overall, INSGENEL achieves the state-of-the-art micro F1 score across eight evaluation datasets with +3.0 relative gain compared with the previous best of discriminative peers, with +6.8 compared with the previous best of generative peers.

The performance of INSGENEL-R is marginally affected since the top-$K$ retrieved entities may not always cover the gold entity. The influence of $k$ on INSGENEL-R's performance is discussed comprehensively in Section 4.3.

Owing to the API quota budget, we only present the INSGENEL-ICL performance of selected four test sets under two OpenAI endpoints in Table 2. The evaluation on `code-davinci-002` and `text-davinci-003` are similar on average, despite varying metrics across different datasets. While our In-Context Learning approach for EL has undergone considerable prompt optimization, it still falls significantly short when compared to our INSGENEL-R which also takes the document and top-$K$ entities as inputs. This may suggest that In-Context Learning for EL needs further investigation and we leave it as future work and list possible solutions in the Limitation Section.

| Method | AIDA | MSNBC | K50 | R500 | Avg |
|---|---|---|---|---|---|
| INSGENEL-ICL | - | - | - | - | - |
| - text-davinci-003 | 50.0 | 53.3 | 39.2 | 34.9 | 44.4 |
| - code-davinci-002 | 60.7 | 47.4 | 39.0 | 25.4 | 43.1 |
| INSGENEL-R | 80.6 | 74.2 | 71.9 | 45.5 | 68.1 |

Table 2: *InKB* Micro $F_1$ reported on selected four test sets . Metrics for INSGENEL-R are listed in the last row for direct comparison.

### 4.3 Ablation Study

We discuss ablation studies on INSGENEL, mainly focusing on the training data efficiency and model size. Additionally, we will evaluate INSGENEL-R retriever top-$k$ gold entity coverage and its influence on the performance of our retrieval-augmented generative EL system.

**Data Efficiency.** Our ablations commence with data efficiency to highlight the superiority of our approach in terms of training data utilization. As depicted in Figure 2, we illustrate the correlation among training data relative size, training compute and EL evaluation performance. The legends indicate that colors of data points represent different EL methods, while the size of data points denotes GPU

hours used for training. INSGENEL, the generative state-of-the-art peer GENRE (Cao et al., 2021b), and the discriminative best model ReFinED (Ayoola et al., 2022) were all trained using V100 GPUs, thus, their training GPU hours are comparable.

We set the training of GENRE using all Wikipedia abstracts as a data size reference point (*i.e.*, a training data ratio of 1) and sequentially downsample all Wikipedia abstract data using coefficients of 0.01, 0.05, 0.1, 0.2, and 0.5 as our comparative data splits. Meanwhile, ReFinED trained on the full volume of Wikipedia, approximately ten times the volume of Wikipedia abstracts. Our best-performing model, trained for around 2.5 days on an 8-V100 node using half of the Wikipedia abstracts, corresponds to 480 GPU hours in the legend. For comparison, GENRE was trained for 30 hours on 64 V100s, and ReFinED for 24 hours on 8 V100s, corresponding to 1,920 and 192 GPU hours in the legend, respectively.

Compared to the previous Generative EL method GENRE, our method exceeded the evaluation performance of GENRE (60.16 vs. 58.2) using just a tenth of the data and a twentieth of the training compute (96 GPU hours vs. 1920). This gap further increased to +6.8 F1 points with the increase of training data and computation.

Likewise, against the earlier Discriminative EL method ReFinED, our method accomplished superior performance (63.72 vs. 61.90) using the same training compute but only 2% of the data volume. Similarly, this lead widened to +3.0 F1 points as training resources increased.

**Model Size.** We seek to explore the potential correlation between model size, type and EL performance by training on different scales of decoder-only generative models. As shown in Figure 3, the five data points correspond to the models of the OPT series 350m, 1.3b, 2.7b, 6.7b, and LLaMA 7b, and their evaluation results after training on the same split of data. We observed a certain emergent ability in the models, with `opt-2.7b` surpassing the previous Generative EL method. Also, despite a similar number of parameters, `opt-6.7b` and `llama-7b` exhibit a noticeable performance gap. This further highlights the ability of our instruction-tuning method to exploit the excellent pre-training quality of LLaMA, as well as to stimulate the latent knowledge transfer capability.

**Retriever Coverage.** Although INSGENEL-R delivers exceptional results in both runtime perfor-

| Method | k | Recall@$k$ (%) | Micro F1 Score (%) |
|---|---|---|---|
| | 5 | 42.57 | 37.70 |
| | 10 | 54.05 | 49.33 |
| INSGENEL-R | 20 | 62.16 | 60.58 |
| | 50 | 75.00 | 67.71 |
| | 100 | 89.20 | 71.90 |

Table 3: Retriever coverage and performance impact of INSGENEL-R's $k$ on K50 test set.

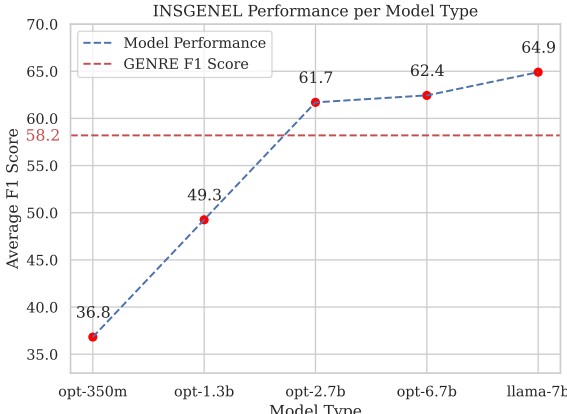

Figure 3: INSGENEL performance with different base models.

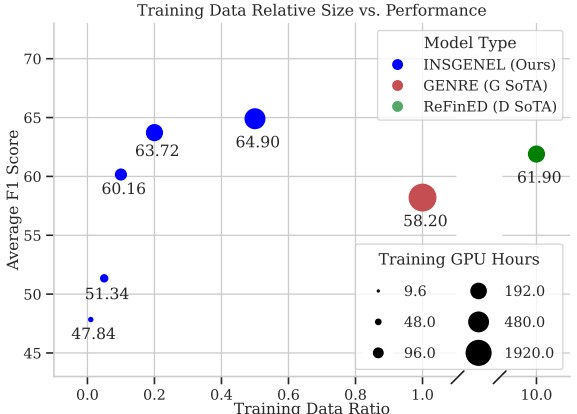

Figure 2: Comparison among training data relative size, training compute and EL performance. Selected works are all trained with V100, thus their training GPU hours are comparable. Letters "G" and "D" in the legend represent generative and discriminative respectively.

| Method | # of Forwards | Runtime (s) | K50 F1 |
|---|---|---|---|
| Ayoola et al. (2022) | 50 | 2.97 | 64.7 |
| van Hulst et al. (2020) | - | 7.32 | 50.7 |
| Zhang et al. (2022b) | - | 131.32$^{\dagger}$ | 64.5 |
| Cao et al. (2021b) | - | 196.30 | 60.7 |
| INSGENEL | 2221 | 160.86$_{\pm 0.52}$ | 73.8 |
| INSGENEL-R | 594 | 44.92$_{\pm 0.28}$ | 71.9 |
| - w/ FA | 594 | 23.76$^{\ddagger}$ | 71.9 |
| - w/ FA + KV | 594 | 16.32$^{\ddagger}$ | 71.9 |

Table 4: Runtime performance benchmark on K50 test set. $^{\dagger}$ denotes that the runtime is estimated based on the maximum throughput of the base model, and the actual runtime should be higher. $^{\ddagger}$ denotes the runtime is estimated based on the typical speedup ratio reported here, and the real runtime may vary. "FA" and "KV" mean FlashAttention and KV caching, respectively.

mance and linking metrics compared with previous generative EL works, one might be curious about how the coverage of entity retriever might impact the EL evaluation results of INSGENEL-R. After all, if the gold entity is not retrieved, INSGENEL-R would be impossible to link any mention to the correct entity. Table 3 reveals the relationship among the number of top retrieved entities $k$, the corresponding gold entity recall@$k$ within the document chunk of length $L = 32$, and the Micro F1 score evaluated on the K50 dataset when completing retrieval-augmented generative EL using the corresponding $k$ entities.

We notice that INSGENEL-R performance generally improves as $k$ increases, which aligns with our intuition since as k increases, candidates cover more gold entities so the chance for INSGENEL-R to link the correct entity also increases. Unfortunately, as the EL checkpoint in Cao et al. (2021b) is not publicly available, we are unable to test whether our retriever-augmented EL scheme would work in other sequence-to-sequence EL frameworks.

## 4.4 Runtime Performance Benchmark

One major barrier to the application of generative EL is its autoregressive nature, which hinders real-time use. In Table 4 we report the runtime of leading and competitive EL systems on the K50 test set covering 50 documents. Among these, our INSGENEL and INSGENEL-R were run 10 times using different random seeds and reported the mean and standard deviation of runtime. Evidently, our model substantially curtails nearly three-quarters of the generative model calls, albeit at a minor sacrifice in accuracy.

Admittedly, there is still nearly $15\times$ the runtime difference compared to the efficiency-centric peer, but we recognize an abundant scope for runtime improvement. For instance, by simply hot-patching attention layers with FlashAttention (Dao et al., 2022), we gain a doubling of inference speed. Also, the decoder-only property of our model enables the

convenience of caching previously generated hidden states, known as KV caching. Furthermore, our retrieval-augmented framework can benefit from parallel decoding with reference (Yang et al., 2023) since many tokens are copied rather than generated – a convenience other discriminative models can not avail. We leave further inference optimization of generative EL as future work.

## 5   Conclusion

We present three variations of generative entity linking (EL) solutions, where INSGENEL realizes an improvement of +3.0 F1 points on popular EL test sets (+6.8 increase compared to previous generative methods). Built upon this model, we propose a novel retrieval-augmented generative EL framework INSGENEL-R that determines when to invoke the large language decoder on the fly. Moreover, our pioneering INSGENEL-ICL marks the inception of in-context learning in EL, despite necessitating additional follow-up research to achieve competitive results.

## Limitations

Although our work pushes the generative EL paradigm to its limit and uses fewer computational resources and training time than most peers, its runtime performance still lags behind that of discriminative EL models, even with a novel retrieval-augmented EL framework in place. This may render our approach suitable for scenarios prioritizing higher linking accuracy over real-time performance. Also, we do not investigate numerous works that improve LM training efficiency, such as low-rank adaption (Hu et al., 2022). These possibilities remain as future work as they could potentially accelerate the training further. In addition, due to budget limitations and inaccessibility to the `gpt-4-32k` endpoint, our INSGENEL-ICL paradigm has not been tested on the GPT-4 series. We may observe a significant performance improvement on the `gpt-4` or `gpt-4-32k`, especially with the latter one's expanded context window that will allow more diverse in-context demonstrations. Last, how to properly organize, select and format exemplars for EL could be an interesting future work of our INSGENEL-ICL paradigm.

## Ethics Statement

Large foundation models carry inherent risks and potential harms, such as the generation of harmful, offensive, or biased content. Even though our work involves controlled generation, we can not guarantee that the fine-tuned model will strictly adhere to ethical requirements in an unconstrained generation setting. As such, we do not recommend a direct conversion from our work to downstream applications without further investigation and mitigation of risks.

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

# A Experimental Details

| Hyperparameter | OPT | LLaMA |
|---|---|---|
| learning rate | 9.65e-6 | 2e-5 |
| weight_decay | 0 | |
| batch size per device | 4 | 3 |
| effective batch size | 128 | 96 |
| learning rate strategy | WarmupLinearDecay | |
| optimizer | AdamW | |
| dropout | 0.1 | |
| gradient clipping | 1.0 | Disabled |

Table 5: Hyperparameter settings for OPT and LLaMA training of INSGENEL.

We implemented all of our neural models using the `transformers` (Wolf et al., 2020) library. As decision spans in INSGENEL-R are not lengthy, we do not notice significant performance improvement in beam search, thus we employ greedy decoding in all INSGENEL-R experiments. We use `beam_size=2` in INSGENEL experiments. If out-of-memory is triggered during training for some model types, optimizer partitioning, gradient state partitioning, and parameter partitioning will be sequentially enabled to ensure successful training completion.

Due to the license attached to LLaMA (Touvron et al., 2023) models, we are not able to directly distribute LLaMA-based weights. Instead, we provide delta weights of our pre-trained checkpoint, and interested readers should fill this form to get base model from Meta AI, then apply delta weights on it to get a functional generative EL model. Hyperparameters for training neural models are listed in Table 5.

Experiments with online OpenAI[5] models were conducted earlier in March 2023 with `openai-python` library. We set `temperature=0` in OpenAI generation to ensure maximum reproducibility, but generation results may differ as OpenAI backend models keep evolving. `gpt-3.5-turbo` endpoint which supports ChatGPT can not correctly adhere to our instruction. For `code-davinci-002` and `text-davinci-003`, we apply generation configuration of max_token=300, top_p=1, frequency_penalty=0.0 and presence_penalty=0.0. We do not run experiments on `text-ada-001`, `text-babbage-001` and `text-curie-001` as they

are with 2,048 tokens of context window, which can hardly satisfy our requirements since a typical length of our input prompt is around 3,000 tokens.

# B Inherent Problems of Vanilla Generative EL

To ensure the legality of generated sequence, *i.e.*, the generated entities are within the KB, vanilla generative EL employs a constrained beam search strategy for inference. At each generation time step, the vanilla approach either chooses to generate the input document verbatim, or start a new mention. Note that this mention start decision is mandatory for each possible token in the document, thus resulting in massive inference overhead as a considerable number of document tokens are unlikely mentioned. See case (a) with grey background in Figure 4, where each arrow or group of arrows represents a forward pass of the generative model.

Once it chooses to start a mention, the vanilla approach seeks advice from a pre-generated prefix tree (a.k.a. trie) to constrain the tokens that are allowed in the next time step and eliminate other options. The same strategy is also used to guide entity identifier generation to remove impossible entities from candidate sequences. Vanilla generative EL relies solely on unique entity title identifiers to distinguish between different entities. This might lead to potential confusion among entities with closely related names when finer-grained information, such as entity descriptions, is not taken into account.

Considering the local optimality of greedy decoding and the potential for it to get stuck in infeasible options during generation, the vanilla approach uses beam search to maintain top-$k$ sequences, falls back to previous states when walking into unreasonable paths, and ultimately parses the top-1 sequence into EL result. As the generative model decides only ONE next step in each beam, some generated sequence in a beam may be illegal to form a valid mention, resulting in wasted inference compute. See case (b) in Figure 4 for details.

Last, the large generative model pre-trained on web-scale text data sometimes suffers from missing important mentions in the generative EL setting even after fine-tuning, which leads to a low recall score during evaluation. See case (c) in Figure 4.

[5] https://platform.openai.com/

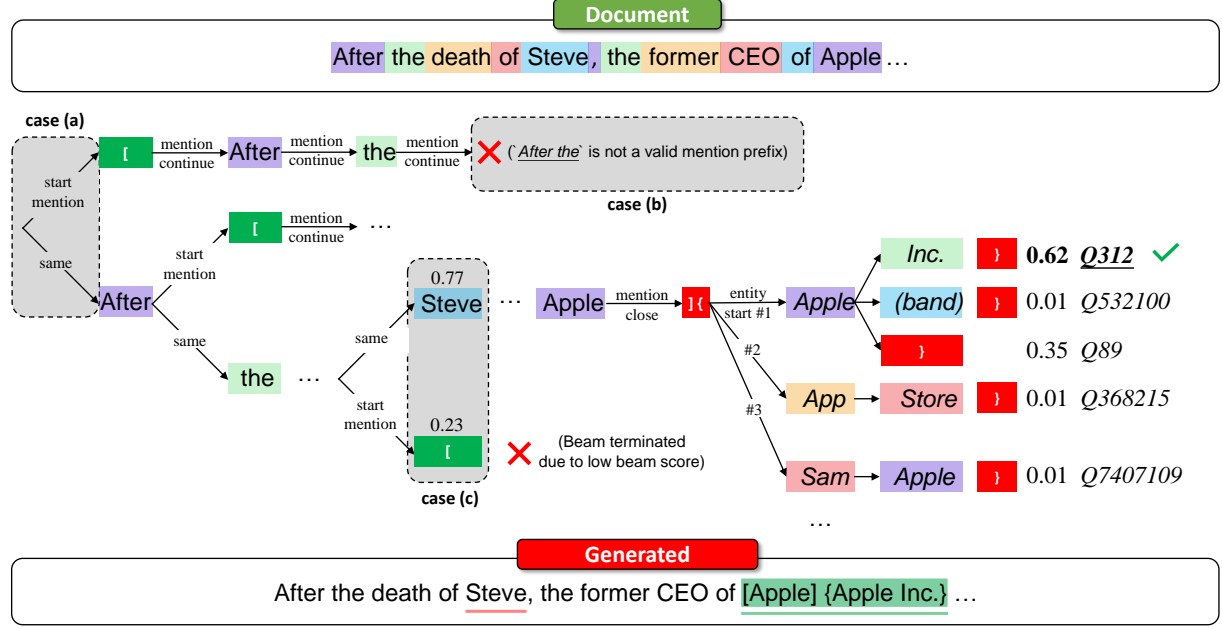

Figure 4: Overview of vanilla generative EL proposed in Cao et al. (2021b) with constrained beam search. Each path is a possible beam during beam search, and the decimal number at the end of each path is the normalized beam score. The decimal numbers in case (c) indicate generative models sometimes make mistakes when initiating mention boundaries. The color scheme for text tokens is adopted from the OpenAI Tokenization webpage and for idea depiction only, and the real tokenization depends on the base model. Best viewed in color.

## C   Entity Linking Setting

### C.1   Candidate Set Construction

Previous neural entity linking systems pre-selects a reasonable number of candidate entities for each mention based on empirical probabilistic $p(e|m)$ scores. We build such a mention candidate dictionary with the combination of Kolitsas et al. (2018) and Hoffart et al. (2011) for INSGENEL, following these systems. Given the generative character of our work, as opposed to a discriminative one, the size of the candidate set does not affect the inference speed. Thus in line with Cao et al. (2021a), we place no restrictions on the candidate set size during inference.

### C.2   Candidate-to-Mention Mapping Construction

Differing from all previous entity linking works, we need an empirical candidate-to-mention mapping for determining the decision range of INSGENEL-R. We reverse the key-value pairs in the mention candidate dictionary in Appendix C.1, remove duplicates of mentions under the same entity entry, and eliminate stop words. The candidate-to-mention mapping, developed from a known dic-

tionary that is utilized by peer works, without the addition of other knowledge or data, provides a fair point of comparison.

### C.3   Evaluation Dataset Statitics

Following Ayoola et al. (2022), we present the topic, number of documents and number of mentions for each dataset used for evaluation. The datasets cover a variety of sources including Wikipedia text, news articles, web text, and tweets. Note that the performance of the model outside these domains may be significantly different.

|  | **Topic** | **Num docs** | **Num Mentions** |
|---|---|---|---|
| **AIDA** | news | 231 | 4464 |
| **MSNBC** | news | 20 | 656 |
| **DER** | tweets | 182 | 242 |
| **K50** | mixed | 50 | 145 |
| **R128** | news | 128 | 638 |
| **R500** | news | 500 | 530 |
| **OKE15** | wikipedia | 199 | 1017 |
| **OKE16** | wikipedia | 254 | 1402 |

Table 6: Dataset statistics for EL datasets

### C.4 Algorithm for Merging Overlapping Decision Spans for Different Mentions

---

**Algorithm 1** Merging Overlapping Decision Span

---

1: **procedure** MERGE(decision_span_lst)
2:     Sort decision_span_lst in ascending order by start
3:     merged_mentions ← []
4:     **for** each tuple (start, end, mention) in decision_span_lst **do**
5:         **if** merged_mentions is not empty and merged_mentions[-1][0] $\leq$ start $\leq$ merged_mentions[-1][1] **then**
6:         merged_mentions[-1][1] ← max(merged_mentions[-1][1], end)
7:         append mention to merged_mentions[-1][2]
8:         **else**
9:         append (start, end, [mention]) to merged_mentions
10:     **return** merged_mentions

---

### Your task is to read the example document and identify the mentions for each candidate entity. Then you need to link these mentions into entities given the candidate entity list. The output is a list of the linked answers and their corresponding mention text. When you are performing this task, remember:

### Rules

- rule 0: Do not include any predictions that do not appear in the document.
- rule 1: Do not make duplicated predictions on the same mention span, unless there are indeed multiple mention texts in the same surface form.
- rule 2: To help disambiguate mentions in the same surface form, you must produce a few surrounding words of the mention in the document, using brackets [] to indicate the start and end of the mention. Predictions without brackets will be considered invalid.
- rule 3: You can not link multiple mentions in a single predicted line. For example, a prediction of `- Linked answer 0: Yosemite National Park; corresponding mention text: [reservoir] near [Yosemite National Park]` is not allowed, because it predicted two mention in a single prediction; instead, you should predict `- Linked answer 0: Yosemite National Park; corresponding mention text: reservoir near [Yosemite National Park]`.
- rule 4: You should locate the mention accurately. For example, `Linked answer 0: United Kingdom; corresponding mention text: [british] government warned friday` would be an acceptable prediction as opposed to `Linked answer 0: United Kingdom; corresponding mention text: british [government] warned friday`.

### Examples

## Example Document 0
began transmitting from Munich, Germany, in 1951, spreading uncensored news to Soviet - controlled countries behind the Iron Curtain during the Cold War

## Example Document 0 Candidates
- candidate 0 for example document 0: Soviet Union
- candidate 1 for example document 0: East Germany
- candidate 2 for example document 0: West Germany
- candidate 3 for example document 0: Iron Curtain
- candidate 4 for example document 0: Europe
...
- candidate 94 for example document 0: Cold War (1962–1979)
- candidate 95 for example document 0: Communist party
- candidate 96 for example document 0: World
- candidate 97 for example document 0: 1972 Summer Olympics
- candidate 98 for example document 0: Kazakhstan
- candidate 99 for example document 0: Cold war (general term)

## Example Answers 0
- Linked answer 0: Munich; corresponding mention text: transmitting from [Munich, Germany], in 1951, spreading
- Linked answer 1: Soviet Union; corresponding mention text: uncensored news to [Soviet] - controlled countries behind
- Linked answer 2: Iron Curtain; corresponding mention text: countries behind the [Iron Curtain] during the Cold War
- Linked answer 3: Cold War; corresponding mention text: Curtain during the [Cold War]

## Test Document
After the death of Steve, the former CEO of Apple, his commencement speech at Stanford was watched thousands of times.

## Test Document Candidates
- candidate 0 for test document: Stanford University
- candidate 1 for test document: Apple Inc.
- candidate 2 for test document: Allen Stanford
- candidate 3 for test document: Stanford, California
- candidate 4 for test document: Apple
...
- candidate 94 for test document: White House
- candidate 95 for test document: Stanford, New York
- candidate 96 for test document: Jesus
- candidate 97 for test document: 20th century
- candidate 98 for test document: 2000
- candidate 99 for test document: Hollywood

## Answers
--------response-------
- Linked answer 0: Steve Jobs; corresponding mention text: death of [Steve], the former CEO of Apple
- Linked answer 1: Stanford University; corresponding mention text: commencement speech at [Stanford] was watched thousands of times

Figure 5: Example prompt input and `text-davinci-003`'s response for an example document from kore50 dataset (Hoffart et al., 2012b). The ellipsis omits the majority of top-100 potential entities for clear depiction. Markdown format highlight is enabled. INSGENEL-ICL with `text-davinci-003` made two predictions in this document, and both were correct. However, it missed an obvious mention "Apple". Best viewed in color.