# OpenReview forum: "Instructed Language Models with Retrievers Are Powerful Entity Linkers"
_EMNLP/2023/Conference — EMNLP 2023 Main_

### Official Review · Reviewer_zVxi · 2023-07-27

**Soundness:** 4

**Excitement:**

3: Ambivalent: It has merits (e.g., it reports state-of-the-art results, the idea is nice), but there are key weaknesses (e.g., it describes incremental work), and it can significantly benefit from another round of revision. However, I won't object to accepting it if my co-reviewers champion it.

**Paper Topic And Main Contributions:**

The paper introduces InsGenEl, a new entity linking approach that uses generative large language models (LLMs) instead of discriminative methods. The authors describe three variations of their method: InsGenEl, InsGenEl-R (a generative approach that also uses entity candidate neural retrieval), and InsGenEl-ICL (which uses in-context learning for entity linking with LLMs).

They also propose a new sequence-to-sequence training entity linking objective with instruction tuning, as well as a new approach for the decoding algorithm of the generative model that uses retrieved information.

The evaluation was performed on multiple entity linking benchmarks, and the results showed that InsGenEl achieved an average improvement of over +3.0 F1 points against discriminative baselines and +6.8 F1 points against generative baselines. Additionally, InsGenEl-R was shown to be 4x faster than the baseline without any loss in quality.

**Questions For The Authors:**

Question A: (Lines 091:094) Could this improvement be solely due to the model's size? Have you tried fine-tuning BART with your approach? Or have you tried using their fine-tuning on your models?

Question B: The paper mentions multiple times instruction-tuning. How does it differ in this case from just fine tuning? The instruction stays the same, it's just a different document that is put in the prompt/prefix/input?

Question C: (Section 3.1) An example of input/target should be provided. The paper should be understood without appendices.

Question D: Instruction: Can you please explain the difference between Vanilla Generative EL and Instruction-tuned InsGenEL? The description in Section 3.1 and 3.2 is not very clear. Is it that Vanilla Generative EL only generates linked entities, while InsGenEL generates the whole document?

Question E: (Lines 275): What's the text of this instruction? Why is it optional? When is it used and when it's not?

Question F: 338:343 How do you deal with not perfect overlap of mentions (due to hyphens, capitalization, inflection)?

Question G: 358:361 - What's the value of this score? How was it chosen? Why is it a reward (is there any RL used)?

Question H: How is token copying performed in the InsGenEl-R? You still have to run the forward pass for those tokens to have key/values for the attention for next tokens, which wouldn't give you any speedup.

Question I: (Section 3.4) It's not possible to understand this part without going to the Appendix ("As a general rule, the paper should be readable as a standalone document, and any details important for understanding the key aspects of the work should be in the paper rather than in appendices").

Question J: (Lines 426:428) - How lack of the overlap between train (Wikipedia 2023-02-20 dump) and benchmarks is ensured?

Question K: (Lines 465:467) - Have you tried to finetune your model on AIDA?

**Reasons To Accept:**

- Introduces novel methods that use pretrained generative large language models.
 - Proposes an approach (InsGenEl-R) that is 4x faster without sacrificing quality.
 - Provides comprehensive evaluation and comparison against discriminative and generative baselines.

**Reasons To Reject:**

- Paper's clarity. Many details and examples are either omitted or put in the Appendix, making it difficult to understand (see questions).
 - The motivation for using generative models rather than discriminative models is not clearly stated.

**Reproducibility:**

4: Could mostly reproduce the results, but there may be some variation because of sample variance or minor variations in their interpretation of the protocol or method.

**Reviewer Confidence:**

3: Pretty sure, but there's a chance I missed something. Although I have a good feel for this area in general, I did not carefully check the paper's details, e.g., the math, experimental design, or novelty.

**Typos Grammar Style And Presentation Improvements:**

line 042: why dense? it could also be used by a sparse network.

line 081: why diverges?

---

> ### Author Rebuttal · Authors · 2023-08-26
>
> We sincerely thank the reviewer for their insightful comments and valuable suggestions. We address each of the concerns below, with particular emphasis on clarifying questions related to the **motivation** of our work.
>
> ### Motivation of our work
>
> We aim to clarify the motivation of our work from two aspects ('why generative' and 'why our generative') to address the motivation concerns about using generative models rather than discriminative ones. We will emphasize this part in the introduction of our future manuscript.
>
> A. Why generative:
>
> Cao et al. [1] were the pioneers in applying prefix-tree constraints for accomplishing generative entity linking. In contrast to discriminative EL methods, which usually treat candidate entities as discrete labels, the autoregressive feature of generative EL methods enrich the interactions between document context and entities. Additionally, prefix-tree constraints in the generative process help to ensure the legitimacy of the entities being predicted.
>
> B. Why our generative:
>
> While generative Entity Linking (EL) methods have many stunning advantages, they are still plagued by several hard-to-ignore drawbacks, as listed below: a) The non-parallelizable beam search decoding makes generative EL extremely slow; b) The performance no longer holds a breakthrough advantage and is even surpassed by subsequent score-based discriminative methods like ReFinED [2] that incorporate knowledge priors, both in terms of metrics and speed; c) It struggles with short and information-scarce entity identifiers since the entity identifiers are the only source a vanilla generative entity linker can see.
>
> In response to these issues, **`INSGENEL-R`** offers a compact solution: By employing a simple vector retrieval, it takes into account the entity description, thereby alleviating the problem of information-scarce entity identifiers. Moreover, the retrieved entities can be utilized in Guided Entity Linking Generation (Figure 1), thereby increasing computational efficiency (fewer model forward passes) and storage efficiency (no longer relying on a large prefix tree, but dynamically generating it during inference). We would like to note that these design features distinguish our method from previous generative EL solutions, which is appreciated by Reviewer hkkb and 7xon. Additionally, our approach can also be viewed as an agent that interacts with the environment (Line 200-206), where documents and retrieved entities are considered as the environment.
>
>
>
> ### Clarity of our work
>
> We apologize for not describing all the important details in the main text, especially regarding the implementation of **`INSGENEL-ICL`**. In light of your comments, we plan to make the following changes in the future manuscript to enhance readability:
>
> A. We will integrate the content of Appendix C into Section 3.4 *INSGENEL-ICL: In-Context Learning Entity Linking Paradigm*, to demonstrate that accomplishing EL tasks using our ICL strategy on a general LLM is still challenging. Not mentioning the specific ICL design in the main text was a significant oversight in our submission.
>
> B. A pair of example input/output will be added to Section 3.1 *Vanilla Generative EL*, to more intuitively reflect the sequence-to-sequence training objective.
>
> C. We will horizontally compare the vanilla generative EL in Appendix Figure 4 with Figure 1, to intuitively show the differences in the implementation methods between the two.
>
> D. The entire Section 3 *Methodology* will be reorganized, and details that significantly contribute to readability from the appendix will be incorporated wherever possible.
>
>
>
> ### Responses for all questions
>
> Question A: (Lines 091:094) Could this improvement be solely due to the model's size? Have you tried fine-tuning BART with your approach? Or have you tried using their fine-tuning on your models?
>
> Answer A:
>
> We do not think that the improvement solely comes from the model size.
>
> 1. LLaMA-7b and OPT-6.7b, which have similar model sizes, also display a noticeable difference (Line 549-555). This suggests that the quality of pre-training, as well as inherent attributes of the model itself (e.g., tokenizer choice, pre-training corpus and domain, model structure, etc.), can also impact the performance of generative EL.
>
> 2. BART has an encoder-decoder structure, whereas LLaMA/OPT are decoder-only. We have not provided a direct comparison, but we did attempt preliminary experiments to perform instruction-tuning on the BART model using the same training data pairs and settings but did not achieve reasonable results. A point of reference might be the average F1 score of 58.2 achieved by Cao et al. [1] using BART-large in a machine translation setting (not able to reproduce it due to possible wrong checkpoint the authors provided).
>
>    You can also refer to our response for Reviewer 7xon for baseline model sizes and corresponding performance metrics of various methods.
>
> ------
>
> Question B: The paper mentions multiple times instruction-tuning. How does it differ in this case from just fine tuning? The instruction stays the same, it's just a different document that is put in the prompt/prefix/input?
>
> Answer B: You are correct; as can be seen from Line 222-230, Entity Linking is a very specific task rather than a zero-shot language task. During the training process, if we consider the input document as part of the instruction, then the entire fine-tuning process can be viewed as instruction fine-tuning. Otherwise, our training process may be seen as CLM.
>
> Furthermore, unlike direct CLM fine-tuning, since the input document is considered a part of the prompt, the contextual relationships within the document itself are not modeled as part of the loss (Line 287-289), as we did not intent to train a general language model.
>
> ------
>
> Question C: (Section 3.1) An example of input/target should be provided. The paper should be understood without appendices.
>
> Answer C: Although we have elaborated on the changes we prepare to make in the part B of 'Clarity of our work' section, we hope to resolve your queries here by providing a toy example:
>
> > For each source provided, write an output that includes the mentions that linked to the appropriate entity identifier. Use brace brackets { } to denote a mention, and box brackets [ ] to denote a linked entity: `[SEP]` After the death of Steve, the former CEO of Apple, his commencement speech at Stanford was watched thousands of times. `[SEP]` After the death of { Steve } [ Steve Jobs ] , the former CEO of { Apple } , his commencement speech at { Stanford } [ Stanford University ] was watched thousands of times. `[SEP]`
>
> ------
>
> Question D: Instruction: Can you please explain the difference between Vanilla Generative EL and Instruction-tuned InsGenEL? The description in Section 3.1 and 3.2 is not very clear. Is it that Vanilla Generative EL only generates linked entities, while InsGenEL generates the whole document?
>
> Answer D: Both vanilla generative EL and InsGenEL generate the whole document during their EL inference.
>
> As Cao et al. [1] stands on the shoulders of encoder-decoder BART, sequence-to-sequence training is quite straightforward by putting document on encoder side and autoregressively generate tokens on decoder side in a machine translation setting.
>
> As for InsGENEL, the entire decoder-only model runs autoregressively. And the training has slight difference (Answer B) as we do not want the model to focus on language modeling inside documents, but on contextualized information that could contribute to accurate entity grounding and disambiguation.
>
> ------
>
> Question E: (Lines 275): What's the text of this instruction? Why is it optional? When is it used and when it's not?
>
> Answer E: The text of instruction is as follows, which went through automatic prompt optimization. The same instruction is shown in the toy example in Answer C.
>
> ```python
> DEFAULT_INSTRUCTION = "For each source provided, write an output that includes the mentions that linked to the appropriate entity identifier. Use brace brackets { } to denote a mention, and box brackets [ ] to denote a linked entity: "
> ```
>
> We apologize for the issues left in our early writing. In all of our experiments, we have appended instructions before input documents. They are not optional.
>
> ------
>
> Question F: 338:343 How do you deal with not perfect overlap of mentions (due to hyphens, capitalization, inflection)?
>
> Answer F: We are wondering you’re discussing a case where the document contains the text "After the death of Steve..." and potential mentions are ['steve', ...]. Appendix D2 explains the details of how we build possible mention sets. We explore every combination of capitalization and hyphenation for mentions so that the possible mentions will cover most cases.
>
> While we don't account for inflection changes, `INSGENEL-R` can still catch the word "apple" in "These fruits are apples," thanks to a surface form match with `apple`. On the other hand, it might fail to identify "Leaves" in "Leaves are falling" if the set of mentions only features "Leaf", which isn't a surface form match.
>
> ------
>
> Question G: 358:361 - What's the value of this score? How was it chosen? Why is it a reward (is there any RL used)?
>
> We used $ \gamma= $1e-4 as the static elevated score. Though the score was chosen empirically, running cross-validation on held-out wiki validation sets could potentially improve evaluation metrics further.
>
> It is worth noting that a similar threshold design also appears in Zhang et al. [3], where $ \gamma $ is used for filtering spans in the document.
>
> Additionally, this score controls trade-off between precision and recall, where the higher the score is, the more likely mentions are predicted in a decision-required span.
>
> And we apologize for the confusion that the word reward brings, and will use the term elevated score instead. There is no RL involved in our work.
>
> ------
>
> Question H: How is token copying performed in the InsGenEl-R? You still have to run the forward pass for those tokens to have key/values for the attention for next tokens, which wouldn't give you any speedup.
>
> Answer H: Line 347-352 and boxes with grey background in Figure 1 show how the token copy works.
>
> To elaborate, we begin with token-by-token generation in vanilla generative EL, where a forward pass is mandatory for deciding whether to start a mention at each generation step, not to mention that some forwards are wasted due to impossible decoding paths during beam search.
>
> In INSGENEL-R, the agent does not have to forward the model to decide anything when out of a decision-required span, but just copy the corresponding token from the document. It is **TRUE** that all copied tokens still have to produce intermediate activations for later tokens to attend on. However, we significantly reduce decoding time by minimizing the number of forward passes. This is particularly evident when no KV Caching optimization is employed, as decoding time correlates almost linearly with the number of forwards in practice.
>
> In addition, we refer interested reviewers to LLMA [6], a preprint concurrent with our work, which elaborates in detail on how to accelerate decoding by utilizing tokens already generated in the LLM context. Our designs share similarities on an intuitive level.
>
> ------
>
> Question I: (Section 3.4) It's not possible to understand this part without going to the Appendix.
>
> Answer I: We fully recognize the oversight of not detailing INSGENEL-ICL in the main part, and we've laid out steps for improvement in the section A of 'Clarity of our work'.
>
> ------
>
> Question J: (Lines 426:428) - How lack of the overlap between train (Wikipedia 2023-02-20 dump) and benchmarks is ensured?
>
> Answer J: The training dataset setup of our work follows established best practices in the EL field, similar to prior works [1,2,3,4,5] that has also used Wikipedia dumps for training and well-known benchmarks for evaluation. After running text similarity checks between abstract part of Wikipedia 2023-02-20 dump and benchmarks, we did not notice overlaps. This aligns well with data collection descriptions of these benchmarks.
>
> ------
>
> Question K: (Lines 465:467) - Have you tried to finetune your model on AIDA?
>
> Answer K: Recent EL works typically follows a recipe of pre-training on Wikipedia then fine-tuning on AIDA. We have not found a recipe on `INSGENEL` that maintains model performance after fine-tuning on AIDA. Training on AIDA trainset improved performance on AIDA but deteriorated performance on all out-of-domain benchmarks. We are still diagnosing this phenomenon and believe that it may be related to larger models overfitting small training sets due to the absence of hyperparameter optimization.
>
> ### Grammar, Style and Presentations
>
> Line 042: Yes, but in this context, what we wish to emphasize is that the encoder in recent EL work is usually neural-based, so we use the term "dense" to indicate that most EL encoders are based on neural networks. Indeed, sparse bag-of-words models can also be used to retrieve entities for entity linking, but this is beyond the scope of this manuscript.
>
> Line 081: We kindly refer you to our response of Question D for the difference.
>
> ### Reproducibility
>
> We notice that you gave an unusual rating on reproducibility but did not raise any related concerns in the question section. Therefore, we would like to clarify the following matters:
>
> a) All training data comes from the publicly available Wikipedia dump, and the data post-processing methods are detailed in Line 433-440.
>
> b) The training code and model weights will be open-sourced after the review process.
>
> c) For experiments conducted on the OpenAI API, as `code-davinci-002` has been deprecated and `text-davinci-003` is also evolving over time, we decide that the generated text and corresponding entity linking parser code we obtained on `INSGENEL-ICL` in March 2023 will also be made public.
>
>
>
> Thank you for your valuable feedback and we hope these could address most of your concerns!
>
> ### References
>
> [1]: De Cao et al. Autoregressive entity retrieval. ICLR 2021.
>
> [2]: Ayoola et al. ReFinED: An efficient zero-shot-capable approach to end-to-end entity linking. NAACL 2022.
>
> [3]: Zhang et al. ENTQA: Entity Linking as Question Answering. ICLR 2022.
>
> [4]: Hoffart et al. Robust disambiguation of named entities in text. EMNLP 2011.
>
> [5]: Kolitsas et al. End-to-end neural entity linking. CoNLL 2018.
>
> [6]: Yang et al. Inference with reference: Lossless acceleration of large language models. arXiv preprint arXiv:2304.04487, 2023.

---

### Official Review · Reviewer_7xon · 2023-08-12

**Soundness:** 3

**Excitement:**

4: Strong: This paper deepens the understanding of some phenomenon or lowers the barriers to an existing research direction.

**Paper Topic And Main Contributions:**

This paper presents a significant advancement in enhancing the efficiency and effectiveness of generative entity link models. This advancement holds particular significance for large language models, as it substantially diminishes the search space during constraint decoding through the utilization of a prefix tree.

In the context of a given document, the proposed method operates by initially identifying a set of pertinent entities. It dynamically regulates the subsequent token generation process, resulting in a substantial reduction of the search space within the generative entity link model.

**Reasons To Accept:**

1. The paper introduces a novel approach by combining a retriever with a language model to enhance entity linking.
1. The proposed method significantly enhances performance and expedites inference, contributing to practical applicability.
1. The paper stands out through its extensive, meticulously documented experimental results that robustly validate the author's assertions.

**Reasons To Reject:**

1. Lack of Experiment Clarifications: The paper falls short in providing sufficient details regarding the experiment settings. It remains unclear whether the baseline methods in Table 1 employ the same language model as the proposed methods. Clarification in this area is essential for proper evaluation.

2. Insufficient Ablation Study on Retriever: While Table 1 reveals that INSGENEL outperforms INSGENEL-R, the effectiveness of the retriever itself requires more comprehensive exploration. To bolster the study, an ablation analysis on the retriever's impact alone should be included. Additionally, considering other retriever types like BM25 or TF-IDF could offer valuable insights into the method's versatility and generalization.

**Reproducibility:**

4: Could mostly reproduce the results, but there may be some variation because of sample variance or minor variations in their interpretation of the protocol or method.

**Reviewer Confidence:**

3: Pretty sure, but there's a chance I missed something. Although I have a good feel for this area in general, I did not carefully check the paper's details, e.g., the math, experimental design, or novelty.

---

> ### Author Rebuttal · Authors · 2023-08-26
>
> We highly appreciate the reviewers' time and effort in providing positive feedback on our manuscript. And we would like to respond to the concerns raised, especially on experiment settings.
>
> 1. Lack of Experiment Clarifications: We apologize for not clearly specifying the type of language model in the experiment part Table 1. We would like to clarify the underlying model for baselines here and in the future manuscript. Note that early non-neural methods are not listed, and `N/A` denotes there are no avg score for reference.
>
>    | Method                 | Model Type (and Size if available)  | Avg  |
>    | ---------------------- | ----------------------------------- | ---- |
>    | Discriminative         | -                                   | -    |
>    | Kolitsas et al. (2018) | BiLSTM                              | 53.4 |
>    | Zhang et al. (2022b)   | 3 * BERT-large (1.02B)              | 60.5 |
>    | Ayoola et al. (2022)   | RoBERTa-base + Task Heads (154M)    | 61.9 |
>    | Generative             | -                                   | -    |
>    | Cao et al. (2021b)     | BART-large (406M)                   | 58.2 |
>    | Cao et al. (2021a)     | RoBERTa-base + BiLSTM (202M)        | N/A  |
>    | Mrini et al. (2022b)   | BART-large + Task Heads (>406M)     | N/A  |
>    | Ours                   | -                                   | -    |
>    | INSGENEL               | LLaMA-7B (7B)                       | 64.9 |
>    | INSGENEL-R             | LLaMA-7B + BERT-large (7.3B)        | 64.5 |
>    | INSGENEL-ICL           | text-davinci-003 / code-davinci-002 | N/A  |
>
>    It's worth noting that discriminative EL methods and generative EL methods are typically **not comparable** in terms of their model sizes, as they operate in different ways during the inference process. Even within the same category of generative EL methods, their efficiency cannot be determined solely based on model size due to variations in decoding strategies (beam search or greedy, constrained or not, etc.). **What might be more informative** is the runtime presented in Table 4, where we compared the inference times of various representative methods.
>
>    We are grateful that the comment raises the baseline model size clarification issue. While we have made every effort to optimize the runtime of `INSGENEL`, it's hard to deny that there is still an order-of-magnitude difference in runtime compared to discriminative methods.
>
> 2. Insufficient Ablation Study on Retriever:
>
>    Due to the length limitations of the initial submission, we were unable to comprehensively discuss the effectiveness of different retrievers and instead focused on retriever coverage through ablation studies in Lines 556-579. We thank for this comment and will append sparse retriever's ablations in Table 3.
>
>    In preliminary experiments, we used BM25 as a baseline for the retriever, and the validation results showed a Recall @ Top 100 of only **35%**. We speculate that this may be due to the fact that entity identifiers are typically short and have surface form overlaps with many documents, totally disregarding semantic similarity.
>
>    Please note that the inefficiency of sparse retrievers in the top-k recall will propagate errors to the generation stage, as `INSGENEL-R` is unlikely to produce reasonable results from incorrectly retrieved candidate entities. Such an error accumulation makes it difficult for sparse retrievers like BM25 to be effectively integrated into `INSGENEL-R`.
>
> Thank you again for your valuable feedback and we hope that our responses clarify any ambiguities.

---

### Official Review · Reviewer_hkkb · 2023-08-14

**Soundness:** 4

**Excitement:**

4: Strong: This paper deepens the understanding of some phenomenon or lowers the barriers to an existing research direction.

**Paper Topic And Main Contributions:**

This paper explores the instruction-tuned autoregressive language models for the task of entity linking. Besides framing the problem as instruction tuning, this work also equips with an effective retriever for entity mention generation, which significantly reduces the inference time of previous unified methods and achieve 4 times speedup. Experimental results on eight out-of-domain datasets demonstrate the superiority to previous generative methods. Finally the authors mentioned the potential of in-context learning settings for the generative entity linking. Overall, this paper has conducted comprehensive studies for tailing recent generative models to the specific entity linking tasks.


**Questions For The Authors:**

Can you explain a bit about the weakness in the ``Reason to Reject’’? Also the in-context learning setting can be explored in the same text2data way.


**Reasons To Accept:**


1. Strong motivation and effective novel retriever for generative entity linking.

2. Ablation study and analysis are comprehensive and detailed, which well supports the claim and explains the experimental results.

3. Point out some interesting directions and improvements of future entity linking work.


**Reasons To Reject:**

The only weakness is lack of careful instruction design. Actually information extraction tasks like entity linking can be converted to text2data problems, i.e., the input is the sentence and the output is the json object or dict of extracted elements. Many works explore instruction tuning on information extractions. It might not be necessary to decode the whole sentences in the autoregressive way.


**Reproducibility:**

4: Could mostly reproduce the results, but there may be some variation because of sample variance or minor variations in their interpretation of the protocol or method.

**Reviewer Confidence:**

4: Quite sure. I tried to check the important points carefully. It's unlikely, though conceivable, that I missed something that should affect my ratings.

---

> ### Author Rebuttal · Authors · 2023-08-26
>
> Firstly, we appreciate the time and effort you invested in reviewing our work, and are especially grateful for your positive recognition. As we delve into the feedback, we would like to provide responses for your comments.
>
> 1. Instruction Design:
>
>    We recognize the potential of `text2data` paradigm in information extraction tasks, especially that `text2data` can output structured JSON format string with bulletproof generation strategy such as jsonformer [1]. However, one primary reason we chose not to employ this approach is the observed tendency of generative models to be less sensitive to numerical indices during our preliminary experiments. For instance, the output JSON of test sentence "After the death of *Steve*, the former CEO of *Apple*" in Figure 1 could be:
>
>    ```
>     [
>       {"mention": "Steve", "entity": "Steve Jobs", "span": [19, 23]},
>       {"mention": "Apple", "entity": "Apple Inc.", "span": [44, 48]}
>     ]
>    ```
>
>    And we haven't been successful in training language decoder end-to-end using these document-JSON pairs, as the model often captured wrong mention span when being evaluated. Outputting a mention span is crucial for a mention-entity prediction as there are usually multiple matches for the same surface form that resolve to different entities.
>
>    It is worth mentioning that such insensitivity could be mitigated when data, compute and model size scale, but it might be more like a general-purpose knowledge grounding agent [2] than a generative entity linker.
>
>    We expect future works on general generative information retrieval tasks, both via fine-tuning a LM or via constructing in-context templates for LLMs.
>
> 2. Necessity for Generating the Whole Document:
>
>    Like the comment said, generating the whole document is not the only option. Apart from generating character index spans as we do in document-JSON training, we can also parse a unique entity linking prediction through generating context on both sides of a mention (see `rule 2` and the `Answers` part of Figure 5).
>
>    In `INSGENEL` setting, the decoding indeed takes some time since at each generation step, the model has to make decision.
>
>    However, with the proposed mention detection offloading strategy, `INSGENEL-R` only needs to focus on potential mention spans. Although it appears that the entire document has been generated, many tokens in Guided Entity Linking Generation process (Figure 1) are actually **copied** from the document, with almost no computational overhead. Reviewer 7xon raised similar issues and we refer interested reviewers to Answer H of response to Reviewer 7xon.
>
>    Therefore, in the `INSGENEL-R` setting, "generating" the entire document might be a more elegant choice. And to maintain **consistency** throughout our work, we have opted for the approach of "generating" the entire document in this manuscript.
>
> Hope these would address all your concerns of our work! And thanks again for the constructive comments.
>
> Reference:
>
> [1]: Jsonformer: a Github repository, URL redacted due to rebuttal policy
>
> [2]: Xie et al. Unifiedskg: Unifying and multi-tasking structured knowledge grounding with text-to-text language models. EMNLP 2022.

---

### Meta-Review · Area_Chair_f9g4 · 2023-09-18

**Recommendation:** 4

**Metareview:**

This paper proposes a new approach to entity linking using generative models, by combining instruction tuning with a retrieval model. In doing so, they achieve improvements on both speed and linking metrics.

Reviewers were impressed by the experimental design and results ("extensive, meticulously documented experimental results"). The major concerns on soundness were with respect to lack of details regarding the effect of model size, and insufficient ablations. The second of these was resolved in the rebuttal. The first seems to still be somewhat of an issue, but the authors are limited somewhat by what models are available.

Although one reviewer was only ambivalent in terms of excitement, the others were more positive, including the most confident reviewer. The reviewer that currently has an ambivalent rating raised many useful points, and believes that the resulting changes will greatly benefit the paper. No reviewers seemed to think this work would be especially transformative, but all seemed to think it worthy of publication.

---

### Decision · Program_Chairs · 2023-10-07

**Decision:**

Accept-Main

**Comment:**

This paper proposes a new approach to entity linking using generative models, by combining instruction tuning with a retrieval model. In doing so, they achieve improvements on both speed and linking metrics.

Reviewers were impressed by the experimental design and results ("extensive, meticulously documented experimental results"). The major concerns on soundness were with respect to lack of details regarding the effect of model size, and insufficient ablations. The second of these was resolved in the rebuttal. The first seems to still be somewhat of an issue, but the authors are limited somewhat by what models are available.

Although one reviewer was only ambivalent in terms of excitement, the others were more positive, including the most confident reviewer. The reviewer that currently has an ambivalent rating raised many useful points, and believes that the resulting changes will greatly benefit the paper. No reviewers seemed to think this work would be especially transformative, but all seemed to think it worthy of publication.